# *C9ORF72* Gene GGGGCC Hexanucleotide Expansion: A High Clinical Variability from Amyotrophic Lateral Sclerosis to Frontotemporal Dementia

**DOI:** 10.3390/jpm13091396

**Published:** 2023-09-19

**Authors:** Izaro Kortazar-Zubizarreta, Africa Manero-Azua, Juan Afonso-Agüera, Guiomar Perez de Nanclares

**Affiliations:** 1Department of Neurology, Bioaraba Health Research Institute, Araba University Hospital-Txagorritxu, 01009 Vitoria-Gasteiz, Spain; 2Molecular (Epi) Genetics Laboratory, Bioaraba Health Research Institute, Araba University Hospital, 01009 Vitoria-Gasteiz, Spain; africa.manero@bioaraba.org (A.M.-A.); gnanclares@osakidetza.eus (G.P.d.N.); 3Department of Neurology, Central University Hospital of Asturias, 33006 Oviedo, Spain; juanafonso1998@gmail.com

**Keywords:** *C9ORF72* gene, frontotemporal dementia, amyotrophic lateral sclerosis, ALS-FTD

## Abstract

The expanded GGGGCC hexanucleotide repeat (HRE) in the non-coding region of the *C9ORF72* gene (C9ORF72-HRE) is the most common genetic cause of familial forms of amyotrophic lateral sclerosis (ALS), FTD, and concurrent ALS and FTD (ALS-FTD), in addition to contributing to the sporadic forms of these diseases. Both syndromes overlap not only genetically, but also sharing similar clinical and neuropathological findings, being considered as a spectrum. In this paper we describe the clinical-genetic findings in a Basque family with different manifestations within the spectrum, our difficulties in reaching the diagnosis, and a narrative review, carried out as a consequence, of the main features associated with C9ORF72-HRE. Family members underwent a detailed clinical assessment, neurological examination, and genetic analysis by repeat-primed PCR. We studied 10 relatives of a symptomatic carrier of the C9ORF72-HRE expansion. Two of them presented the expansion in the pathological range, one of them was symptomatic whereas the other one remained asymptomatic at 72 years. Given the great intrafamilial clinical variability of C9ORF72-HRE, the characterization of patients and family members with particular clinical and genetic subgroups within ALS and FTD becomes a bottleneck for medication development, in particular for genetically focused medicines for ALS and FTD.

## 1. Introduction

Amyotrophic lateral sclerosis (ALS) and frontotemporal dementia (FTD) are two rare adult-onset heterogeneous neurodegenerative disorders, characterized by motor and behavior syndromes, respectively.

ALS is the most common among motor neuron diseases (MND), representing 80–90% of all cases of MND [1], with an estimated incidence of 1.75–3 per 100,000 and a prevalence of 10–12 per 100,000 people [2,3,4]. The hallmark of ALS is a progressive dysfunction of both upper motor neurons, located in the motor cortex, and lower motor neurons, found in the brainstem and spinal cord [5]. Lower motor neuron signs include muscle wasting (amyotrophy), weakness, and fasciculations, whereas upper motor neuron signs include hyperreflexia (such as clonus, and positive Babinski and Hoffman signs) and spasticity [6]. TDP-43 immunoreactive neuronal cytoplasmic inclusions are the distinctive neuropathological molecular signature present in nearly all cases of sporadic ALS (sALS) and the majority of familial ALS [7]. Most cases of ALS are sporadic, but a causative gene mutation can be found in about 15% of cases (see below) [5]. 

FTD constitutes the cause of approximately 10% of all dementias [8] and is a common cause of dementia of pure neurodegenerative origin in people younger than 65 years [9]. It predominantly affects the frontal and/or temporal lobe and typically results in behavioral changes, speech problems, and executive symptoms. The behavioral variant (bvFTD), and the semantic and non-fluent variants of primary progressive aphasia (svPPA and nfvPPA), are all included under the umbrella term FTD. Disorders linked to FTD may also include neurodegenerative conditions such as corticobasal syndrome (CBS) and progressive supranuclear palsy (PSP), which may exhibit symptoms of frontal lobe dysfunction over the course of their clinical manifestations [10]. The term used to describe the pathology associated with all types of clinical FTD is frontotemporal lobar degeneration (FTLD), which refers to the selective degeneration of the frontal and temporal lobes. There are three underlying pathologies in FTLD: 50% have ubiquitin-positive TAR DNA-binding protein (TDP-43) pathology (FTLD-TDP), 40% have neuronal and glial inclusions of the microtubule (MT)-binding protein tau pathology (FTLD-tau), and 10% have ubiquitinated inclusion bodies immunoreactive for the fused in sarcoma (FUS) pathology (a-FTLD-U) [11]. Population and clinical studies have revealed that 30–50% of cases of FTD are familial, with 10% being transmitted in an autosomal dominant pattern [12,13]. 

Both clinical syndromes have been described within the same family or even within the same individual with a combined phenotype [14,15]. Approximately 50% of ALS patients are categorized as having ALS-FTD, which includes behavioral changes and cognitive abnormalities linked to frontal and temporal lobe degeneration [16]; specifically, 30% of ALS patients present symptoms of bvFTD [17]. FTD-MND is a subtype of FTD that affects about 40% of patients with bvFTD who also have motor neuron dysfunction and meet the criteria for MND [16,18], and 15% of bvFTD cases progress with symptoms of ALS [19]. 

The identification of the phenotypical linkage supported the detection of the genetic association to a locus at chromosomal region 9p13.2–21.3 in genome-wide linkage studies [14,15,20,21,22,23]. In 2011, the heterozygous hexanucleotide GGGGCC >30 replications of the expansion in the non-coding region of the *C9ORF72* gene (C9ORF72-HRE) were described as the cause of both FTD and ALS [24,25]. Such expansion of the hexanucleotide of *C9ORF72*, apart from being the most common genetic cause of ALS and FTD, causes up to 88% of patients with familial ALS-FTD [7].

Some other genetic causes have been associated with ALS [6], these being Cu-Zn superoxide dismutase (*SOD1*) [26], TAR DNA Binding Protein (*TARDBP*) [27], and Fused in Sarcoma (*FUS*) gene [28,29], the most frequent causes after *C9ORF72*. Regarding FTD, other involved genes include the microtubule-associated protein tau (*MAPT*) and progranulin (*GRN*), among others [30].

Currently, the early and definitive diagnosis of both diseases, ALS and FTD, remains challenging because of the limited accuracy of the neuroimaging and molecular biomarkers. In fact, they are considered to constitute an overlapping neurodegenerative syndrome, with patients presenting all along a varied clinical spectrum and a shared molecular pathogenesis [31]. A diagnosis other than ALS or FTD is given to about 35% of patients with *C9ORF72* mutation when the illness first manifests; this occurs as a result of the 35% of patients’ atypical onset presentations, which frequently mimic other neurodegenerative disorders [32]. The aim of this investigation involves a comprehensive description of the genetic and clinical characteristics exhibited by a family with FTD and ALS caused by the *C9ORF72* hexanucleotide expansion and a narrative review of the status of the diagnostic markers of *C9ORF72*, related neurodegenerative diseases.

## 2. Materials and Methods

### 2.1. Characterization of a Large Basque Family with Both ALS and FTLD Cases

Clinical evaluation, neurological examination, and genetic analysis were performed on 11 family members. Each person was tested with a Montreal Cognitive Assessment (MOCA) neuropsychological exam, which has been featured by optimal diagnostics as a screener for cognitive impairment in non-demented ALS patients [33]. 

#### 2.1.1. Sample Collection and DNA Purification

Blood samples from the 11 individuals were collected in EDTA tubes after obtaining the signed informed consent for diagnostic and research purposes.

DNA purification was carried out from 1 mL in a QiaSymphony (Qiagen) automated system.

#### 2.1.2. C9ORF72 Hexanucleotide Expansion Analysis

*C9ORF72* was genotyped using AmplideX^®^ PCR/CE C9ORF72 Kit (Asuragen, Austin, TX, USA), based on a repeat-primed PCR (RP-PCR) approach which accurately sizes fragments up to 145 repeats and is also able to detect expansions longer than that number of repeats. RP-PCR products were analyzed in an ABI3130xl automated analyzer (Applied Biosystems, Foster City, CA, USA) and data analysis was carried out using GeneMapper software (ThermoFisher Scientific, Waltham, MA, USA) with an adapted bin provided by the manufacturer of the kit.

#### 2.1.3. 9p21 Haplotype Construction

Haplotype analysis was performed using four microsatellite markers (from telomeric to centromeric, *D9S169*, *R14866*, *D9S191*, *D9S263*) located at 9p21.2 around the *C9ORF72* gene. Microsatellite typing was performed by fluorescent PCR and the fragments’ lengths were stablished by capillary electrophoresis on an ABI3500 Genetic Analyzer using GeneScan^TM^ 600 LIZ^TM^ Size Standard (ThermoFisher Scientific). 

### 2.2. Narrative Review

To compile the most relevant published evidence on the association of C9ORF72 expansion and ALS and FTD, a comprehensive literature search was conducted using PubMed including articles published from 1 January 1946 through 31 July 2023. The search terms used were “*C9ORF72* gene”, “C9ORF72-HRE”, “frontotemporal dementia”, “amyotrophic lateral sclerosis”, “ALS-FTD”, “ALS”, “FTD”.

We exported the results of each search to a reference management software, which eliminated the duplicates. Only articles published in English and with a clear diagnosis of the pathologies were included. No restrictions were placed on the study design. The total number of records reviewed was 244.

## 3. Results

### 3.1. Characterization of a Large Basque Family with Both ALS and FTLD Cases Related to C9ORF72-HRE

The clinical and genetic results obtained have been reflected in the familial pedigree shown in Figure 1.

#### 3.1.1. Neurological Findings

The main clinical signs are summarized in Table 1.

The index case (II.5), an 80-year-old right-handed man was first examined in 2019, and he underwent multiple reassessments over a year due to a gradual decline in cognitive function and mild behavioral and personality changes since the age of 78. His wife complained that he had severe apathy and that he was emotionally detached. In addition, he was losing his memory over time and he was forgetting things like appointments and the whereabouts of common objects in his house. She also claimed that he was losing his ability to recognize common places. For instance, he frequently became lost in his town. There were reports of visual illusions, such as referring to seeing snakes at his home, and presenting bizarre ritualistic behaviors such as checking under beds and closing all doors before going to bed. There were no alterations in eating habits or food preferences. There were no symptoms of autonomic dysfunction, or insomnia. Nevertheless, he was independent in basic activities of daily living and continued to maintain his hobbies (walking, reading, watching the news, etc.). His symptoms remained constant. The patient’s clinical history for prior psychiatric diagnoses and treatments was unremarkable. There was no prior history of drug or alcohol abuse. 

The initial neurological examination showed severe language impairment, characterized by low category and design fluency, altered naming, monosyllabic responses to simple questions, and incomprehension of some simple commands. The physical neurological examination was normal, with no evidence of parkinsonism or motor neuron signs. He had a score of 20/30 on the Mini-Mental State Examination (MMSE) where orientation, recall, repetition, visuospatial function, and following commands were the penalized points. An attempt was made to perform the MOCA test, but it was not possible due to language impairment (he did not understand the numerical series, could not read the clock, could not name simple animals, etc.).

Cranial magnetic resonance imaging (MRI) showed a bilateral marked frontotemporal cortical atrophy pattern, in addition to cerebellar vermix atrophy. Taking into account imaging results and clinical manifestations, he was diagnosed as presenting bvFTD.

While his symptoms remained stable and his neurological and neuropsychological evaluations remained unchanged over the two years, his wife did observe a slight increase in apathy and executive dysfunction.

The index father (I.2) died around 70 years of age in an accident and he had a long-lived mother (I.1) with no deterioration. One of his sisters (II.6) died with MND. A paternal uncle (I.3) and two of his cousins (II.10 and II.12) also died of MND. In addition, his family history contained an affected elder sister (II.1) with a previous PPA diagnosis (Figure 1). Otherwise, there was no other known family history of psychiatric or neurological disorders.

The sister (II.6) was diagnosed with ALS in 1994, when she was 51, debuting with bulbar onset. She finally died at the age of 69 years, with no genetic material available at the time of the current study. 

We have not been able to access the complete medical history of the rest of the affected patients with MND in the family.

Individual II.1 was an 86-year-old woman, with no previous personal history of interest except arterial hypertension, malaria, and paroxysmal atrial fibrillation, who lives in community as a cloistered nun. Cognitively preserved and with an active life until the date of the first consultation in 2015, when she presented with progressive difficulties in language construction, mainly in the articulation of words, although comprehension was maintained. In addition, she also had failures in writing associated with the presence of paraphasia and reading difficulties, losing the meaning of sentences with isolated comprehension failures.

In the Barcelona Test, she showed less verbal fluency without dysnomia or problems with repetition, comprehension, or writing. 

A MRI was performed, which was normal.

In the follow-up visit in 2021, the patient was found to have an autonomous functional situation and was able to handle money, run errands, organize medication, and walk independently. On examination, the patient was conscious and temporal–spatially oriented, aware of her illness, and cooperative on examination. The physical neurological examination was normal. There was scarce fluency both in the mother tongue, Basque, and in Spanish, but with normal structure, with occasional phonetic paraphasias. Object naming and comprehension were maintained. There were no associated personality changes or behavioral disorders.

She was diagnosed with nfvPPA at age 88.

If we analyze the clinical information available for four affected members, there was wide variation in age of onset (mean = 54.3, range = 34–74 years) and disease duration (mean = 5.3, range = 1–16 years). Our results share previous findings, showing an earlier age of onset and a worse overall prognosis associated with MND-type debut [34]. 

#### 3.1.2. Familial Genetic Test Results

Of the eleven individuals who have participated in the genetic study, the index and two other members (II.8 and II.10) carried C9ORF72-HRE. II.8 was asymptomatic and II-10 had suffered ALS. The remaining eight were not carriers of the expansion. Surprisingly, one patient (II.1) does not carry the expansion even with her diagnosis of nfvPPA. Given this finding, we considered two possibilities: it was either a phenocopy or a reversal of the expansion. To solve this problem, we carried out a microsatellite analysis and subsequent generation of the different haplotypes. Thus, we were able to conclude that this woman did not have the haplotype associated with the disease and, therefore, was a phenocopy. A phenocopy is an individual with clinical features similar to those of a specific disease, but without carrying the underlying genetic cause.

Although it has been described that some patients with PPA carry C9ORF72-HRE, this disease has been classically associated with mutations in the *GRN* and *MAPT* genes. So, further studies are needed on this patient to make a differential diagnosis of Alzheimer’s disease, sporadic FTD, or FTD with different genetic causes (MAPT or progranulin, for example).

We also confirmed that individual II.8, an asymptomatic carrier, shared the affected haplotype with the index (II.5), which indirectly confirms the HRE analysis results. This result reveals a low penetrance for the HRE in this individual.

Although the inheritance associated with *C9ORF72* is autosomal dominant and, therefore, each descendant of a carrier has a 50% chance of inheriting the expansion, in this family, the percentage of carriers is lower than would be expected a priori. A few years ago, French researchers had already suggested the possibility that gender might influence the transmission of the expansion as, in their study, it was more frequent when the carriers were the mothers than when the fathers were the carriers [35]. In this family, the carriers of the expansion are males. Perhaps this may explain the distortion of transmission.

### 3.2. Narrative Review of Neurological Findings of C9ORF72-HRE Patients and Diagnostic Biomarkers’ Role

#### 3.2.1. Clinical and Neurological Examination

A behavioral variant (bvFTD) is observed in the majority of C9ORF72-FTD and C9ORF72-FTD-ALS patients [36], while the semantic variant of primary progressive aphasia (PPA) and the nonfluent/agrammatic variant (nfvPPA) are less commonly observed [37] and a rare case with the logopenic variant of PPA was also documented [38]. In addition, bvFTD related to *C9ORF72* expansion may be indistinguishable from the bvFTD classic variant with apathy, disinhibition, socially inappropriate behavior, abnormal eating patterns, and loss of empathy [19]. Curiously, all these manifestations were observed in the different members of the studied family.

Particularly noticeable are the higher frequencies of psychotic, hallucinatory, and delusional symptoms found in the *C9ORF72* expansion carriers [39], as with the index of the present family, who suffered from visual illusions and presented strange ritual behaviors. Prior to being identified as having FTD, C9ORF72-HRE carriers (especially those without neurological symptoms) may be diagnosed with bipolar disorder [40], schizophrenia [40], obsessive–compulsive disorder [41], and dementia with Lewy bodies [41]. Episodic memory issues emulating Alzheimer’s disease [42,43,44,45], parkinsonism with akinetic-rigid syndrome [46,47,48,49,50], and cerebellar signs can be observed [46,51].

A recognizable phenotype of ALS patients with the *C9ORF72* repeat expansion appears to include earlier disease onset, the presence of cognitive and behavioral impairment, and decreased survival [34]. Recent studies indicate that bulbar onset is significantly more frequent, around 34% compared to sporadic cases, 23% [52]. These findings can be evidenced in our family, as the sister (II.6) suffering from ALS was diagnosed at the age of 51 years due to bulbar deterioration, but she had a long survival and deceased when she was 69. 

Patients with C9ORF72-HRE show abnormal results in memory, verbal fluency, and executive functions during neuropsychological screening, generally performing worse than other *SOD1*-mutation carriers or patients with no genetic variant [52]. 

Patients with C9ORF72-HRE experience movement disorders very frequently. Tremors or parkinsonism are reported in more than 60% of patients; 30–40% of patients have myoclonus, dystonia, and chorea and less than 10% of patients have ataxia [53]. These features were not observed in any of the affected patients in our family.

The Huntington’s disease-like (HDL) syndrome, which resembles Huntington’s disease but lacks the CAG expansion seen in *HTT*, is another intriguing onset phenotype. In particular, C9ORF72-HRE is present in about 5% of patients with HDL syndrome [54]. 

In presymptomatic C9ORF72-HRE carriers, decreased verbal fluency, non-verbal memory, and executive functions have also been observed [55]. These individuals present lower praxis scores and intransitive gesture scores compared to controls even when the analyses were limited to participants under the age of 40 [56]. 

#### 3.2.2. Neuroimaging Biomarkers

Even though there is no particular neuroimaging pattern, unlike in patients with *GRN* or *MAPT* mutations, symmetrical frontotemporal atrophy is the most frequent finding [39,48,57,58,59,60,61,62]. When compared to controls, the MRI imaging of C9ORF72-HRE-patients exhibits typical FTD atrophy, but this atrophy pattern is atypical when compared to other FTD genes [60,61]. Dorsolateral, medial, orbitofrontal, parietal, thalamic, and cerebellar atrophy are more pronounced in FTD-ALS patients [60,61]. In our case, the index patient presented this atrophic pattern bilaterally, in addition to cerebellar vermix atrophy. 

Functional neuroimaging using FDG-PET and SPECT has revealed changes to the frontal, temporal, and parietal cortices and subcortical structures, similar to structural neuroimaging [48,63,64].

Functional connectivity studies connect the atrophy of the left medial pulvinar thalamic nucleus in patients carrying C9ORF72-HRE with decreased salience network connectivity [65].

In presymptomatic C9ORF72-HRE carriers, the thalamus and most other brain regions are smaller than that of controls 10 to 40 years prior to onset [56,66]. MRI-measured brain atrophy is the first biomarker of change in C9ORF72, even before other biomarkers such as NfL change [66]. Up to 10 years before symptoms appeared and before changes in gray matter volume become appreciable, the C9ORF72 repeat expansion is linked to modifications in brain glucose metabolism [67]. These results suggest that in people with this underlying genetic basis for FTD, FDG-PET may be a particularly sensitive and practical method for examining and monitoring the early stages of FTD [67,68].

#### 3.2.3. CSF and Blood Biomarkers

A single study indicates that in an FTD cohort [69], *GRN* mutation carriers had higher levels of NfL and lower p/t-tau ratios than patients with the *C9ORF72* mutation, *MAPT* mutation, or no known mutation. In the C9ORF72 mutation subgroup’s analyses, the NfL levels were highly variable, with many of them having low levels. Both NfL and p/t-tau predicted survival in the analyses of the whole cohort.

There are some studies on poly(GP), poly(GR), and poly(GA) in cerebrospinal fluid as potential biomarkers for C9ORF72-ALS/FTD [70,71,72].

In presymptomatic *C9ORF72* gene carriers, recent findings outline that NfL levels in C9ORF72 started to differ from controls about 30 years before symptoms appeared and remained noticeably higher than controls throughout all presymptomatic epochs [66]. Results from some trials show that plasma NfL might make it easier to find carriers on the verge of phenoconversion [73,74].

Actually, lack of validation across studies has made proteomics for the study of FTD and ALS biofluids still difficult [75].

The pattern of these biomarkers cannot be compared within our family, as we lack these types of studies.

#### 3.2.4. Genetic Aspects

The *C9ORF72* gene includes two non-coding exons (1a and 1b) and ten coding exons (from 2 to 11) and gives rise to three coding variants. Variant 1, V1 (NM_145005), is a short transcript including non-coding exon 1a and exons 2–5 as the coding sequence that generates the 222-amino acid isoform (C9-short of 24 kDa). V2 (NM_018325) encompasses non-coding exon 1b and exons 2–11 as the coding sequence, whereas V3 (NM_001256054) includes non-coding exon 1a and coding exons 2–11. These two variants generate the 481-amino acid isoform (C9-long of 54 kDa) (Figure 2A) [25].

The hexanucleotide repeat is located between exons 1a and 1b. The size of the hexanucleotide repeat expansion mutations (HRE), also known as G4C2 expansions, in healthy subjects ranges between 2 and 24, with most people harboring two to eight repeats. An expansion above 30 is considered pathological; a small proportion of patients present short expansions (30–100 units), while the vast majority have many hundreds or thousands of repeats [24,76]. As in other neurologic disorders caused by expansions, mosaicism caused by the somatic instability of the repeat number in several tissues and among central nervous system regions has been described, leading to a variety of mutation sizes in different tissues with variation within tissues within the same individual [77]. Moreover, it has also been observed that the repeat number varies in blood with age at the collection and over time in successive blood collections from *C9ORF72* mutation carriers [78].

Furthermore, the *C9ORF72* GGGGCC expansion presents an age-dependent penetrance. The variability described in the literature is very wide with individuals who develop the disease in their twenties and a small number of patients who remain asymptomatic at ninety years. For C9ORF72, age-dependent penetrance is estimated as follows: ~0% at age 35, 50% at age 58, and close to 100% at age 80 [79]. Moreover, it seems that the major allele (T) of rs1990622 at *TMEM106B* functions as a modifier, leading to later death and onset ages [80]. All these aspects make it difficult to establish correlations between repeat expansion size, age at onset, and/or disease severity.

The non-coding G4C2 expansion has two main consequences [24]. First, a loss-of-function effect causing C9ORF72 haploinsufficiency that seems to impair synaptic vesicle recycling, cause lysosomal accumulation, alter glial secretomes (contributing to neuroinflammation), and affect autophagy-mediated RNA homeostasis [81,82]. Second, a gain of function is associated with the expression of abnormal bidirectionally transcribed RNAs carrying the repeat [82]. Despite being in a non-coding region of the gene, the expanded hexanucleotide repeats are bidirectionally transcribed into repetitive RNAs via a noncanonical translation mechanism called RAN (repeat-associated non-ATG), generating five different arginine-rich dipeptide repeat (DPR) proteins that have been shown to alter nucleocytoplasmic transport impacting neuronal survival. DPRs with polyglycine-alanine (Gly-Ala) and polyglycine-arginine (Gly-Arg) are translated from the sense strand; polyproline-alanine (Pro-Ala) and poly proline-arginine (Pro-Arg) are translated from the antisense strand; poly Gly-Pro is also coded by both senses [83,84,85,86,87,88]. If not translated, the RNA can be accumulated in RNA foci forming secondary structures (i.e., hairpins, stable G-quadruplexes, DNA–RNA heteroduplexes, and RNA duplexes), which affect promoter activity, genetic instability, RNA splicing, localization, and transport system (through the sequestration of RNA-binding proteins) (Figure 2B) [82].

#### 3.2.5. Neuropathology

The TDP-43 proteinopathy C9ORF72 neuropathology exhibits the characteristic immunoreactive aggregates that are positive for ubiquitin and TDP-43 in neuronal cytoplasm [7]. However, in a few ways, it differs from other TDP-43 proteinopathies. 

In C9ORF72-ALS, a highly specific feature is that the majority of the ubiquitinated neuronal cytoplasmic inclusions and lentiform neuronal intranuclear inclusions are nucleoporin 62 (p62)-positive but TDP-43-negative [89]. p62 is thought to play a role in the movement of proteins and mRNA into and out of the nucleus [63]. 

Dipeptide repeat proteins have been seen in neuronal cytoplasmic and intranuclear inclusions in CNS samples from C9ORF72 cases, with DPRs from the sense strand being more common than dipeptides related to the antisense strand [83,84,85,86,87,90,91,92,93]. As a main feature of the pathology, DPRs do not co-localize with TDP-43, only with p62 [94,95]. DPR aggregates can be seen in the frontal, occipital, temporal, and motor cortex as well as in the subcortical regions, the midbrain, the cerebellum, and the spinal cord [86]. 

The presence of intracellular aggregates of RNA with the expanded repeat is another indicator of C9ORF72 HRE neuropathology [87], and they are found by fluorescent in situ hybridization (FISH) [84]. These RNA foci are found in a variety of cell types, such as astrocytes, microglia, and motor neurons, as well as in the frontal cortex, motor cortex, hippocampus, cerebellum, and spinal cord, as well as in lymphoblasts, fibroblasts, and iPSC-derived neurons [85,86,90]. Cells with abnormalities in the TDP-43 protein appear to accumulate RNA foci [93].

It is unknown whether C9ORF72-ALS, C9ORF72-ALS/FTD, and C9ORF72-FTD have distinct neuropathology, although the presumption is that they do not [94].

#### 3.2.6. Future Therapeutic Strategies

Currently, there is no cure for ALS-FTD brought on by C9ORF72. 

The progress in novel potential therapies for C9ORF72-related ALS and the possibility of the early identification of individuals genetically at risk of developing the disease is generating a critical need for biomarkers to track neurodegeneration that could be used as outcome measures in clinical trials. Asymptomatic carriers of C9ORF72 variants are the best study population to characterize early pathological changes; for this reason, the presymptomatic stage of the illness offers a priceless chance to extend the therapeutic window. Moreover, studies have extensively discussed the presence of pre-clinical changes in carriers of the expansion up to 20–25 years before the expected onset of symptoms. These studies also suggested that very early detectable structural imaging changes, particularly in the C9ORF72 group, occurred more than 20 years before symptoms were expected to appear [96].

Any proposed therapy must take into account the dual mechanism by which C9ORF-HRE generates the disease, both haploinsufficiency and gain of function.

Viral vector-mediated gene therapy is a promising treatment option for genetic degenerative diseases, specifically for C9ORF72 ALS-FTD. The use of viral vectors, such as adeno-associated viral (AAV) vectors with neural tropism, enables the correction of defective genes in affected tissues, replacing a dysfunctional gene or lowering the expression of toxic proteins. To treat C9ORF72-ALSFTD, silencing sequences are being expressed through the use of AAV. The viability of miRNA-based and AAV-delivered gene therapy in experimental models was demonstrated by two studies funded by UniQure. When an AAV serotype 5 (AAV5) expressing synthetic miRNA was administered to ALS mouse models, the number of repeat-containing C9ORF72 transcripts and RNA foci significantly decreased [97,98]. However, there has not yet been a patient clinical trial. 

Some researchers try to avoid using viral vectors because of possible side effects. One promising approach is the use of antisense oligonucleotides (ASOs). Initially, when the involvement of haploinsufficiency in disease development was unknown, different studies developed ASOs that depleted all C9ORF72 transcripts or those selectively containing the expansion [71,91,99,100,101]. Currently, the importance of preserving C9ORF72 protein expression has been emphasized in the development of new ASOs [102], but strategies aimed at restoring control levels of this protein are still lacking. There is currently a clinical trial promoted by Biogen using this methodology (clinicaltrials.gov Identifier: NCT03626012). Specifically, the ASO designed and tested by this company selectively targets *C9ORF72* transcript variants 1 and 3 that carry the expansion.

Other avenues of study have been proposed, such as CRISPR-mediated therapy to generate deletions in the promoter to reduce the expression of the *C9ORF72* variant containing the repeat expansion [103]. These are viable strategies for upcoming in vivo studies.

## 4. Conclusions

This research has allowed performing a detailed genetic and clinical description of a new pedigree of the C9ORF72 mutation, showing the variability with both frontotemporal variant behavioral dementia and cases of amyotrophic lateral sclerosis.

A limitation of our study is the small proportion of carriers. Since the disease is inherited in an autosomal dominant manner, each offspring of a carrier has a 50% chance of inheriting the genetic alteration. A priori, it would have been expected that five or six patients of the second generation presented the expansion, but the detected carriers have been lower than expected. This low number has limited the possibility of evaluating the degree of penetrance of this pedigree.

The present paper aims to highlight the importance of genetic testing of ALS and FTD patients, which will enable the identification of gene mutations both in familial and also in sALS subjects, expanding the possibility of a targeted treatment for these patients along with the development of broader treatments. 

Moreover, the evidence from this study suggests that a paradigm shift is expected in routine clinical practice when dealing with cases of asymptomatic carriers of the pathology. The presymptomatic and early symptomatic phase of C9orf72 carriers has long been a topic of academic interest, but since antisense oligonucleotide therapies are in trial, the assessment of presymptomatic disease burden has begun to have real-world application.

## Figures and Tables

**Figure 1 jpm-13-01396-f001:**
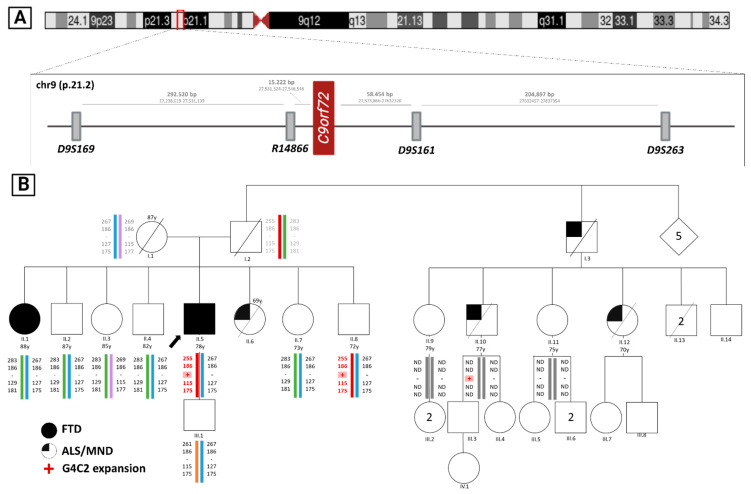
(**A**) Representation of the studied microsatellites’ locations in chromosome 9 (9p21.2) and the distance between them; (**B**) FTD-ALS family pedigree and haplotype analysis. Squares represent males, circles represent females, diamonds represent people of unknown sex, crossed symbols indicate that the person is deceased and the arrow points to the index. Colored symbols indicate FTD diagnosis and one-quarter-colored symbols indicate ALS patients. Below the symbols, the generation is indicated in Roman numerals and the individual in Arabic numerals, and colored bars represent each haplotype, the red bar corresponding to the haplotype carrying the genetic alteration in C9ORF72 (pink shading). The numbers over the symbols indicate the age at decease. In addition, the haplotypes of individuals I.1 and I.2 have been inferred from the descendants due to lack of sample. ND: Microsatellites analysis not performed. Y: age at study (in years). Partially created with BioRender.

**Figure 2 jpm-13-01396-f002:**
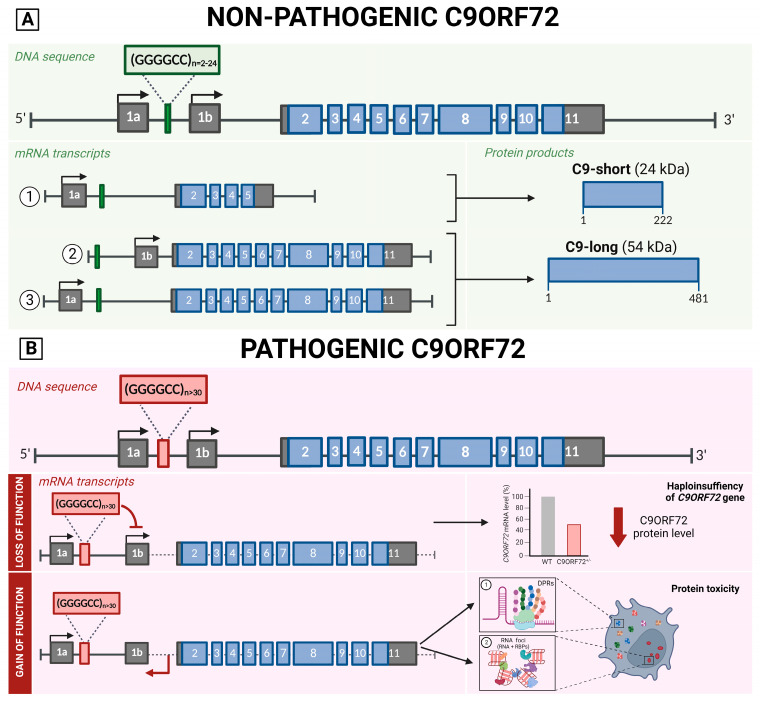
Schematic representation of *C9ORF72* gene structure, its multiple transcripts, and protein isoforms in a non-pathological (**A**) and pathological state (**B**). (**A**) The *C9ORF72* gene sequence contains the hexanucleotide repeat between exons 1a and 1b, in the non-pathological cases the (G4C2)_n_ expansions ranges between 2 and 24. This DNA sequence encodes for three transcript variants. (**B**) In the pathological state, (G4C2)_n_ expansion is above 30 and causes 2 consequences. Loss of function mechanism (Above): The abnormal expansion of G4C2 inhibits the 1b transcription, leading to C9ORF72 haploinsufficiency. Gain of function mechanism (Below): the expanded hexanucleotide repeats are bidirectionally transcribed into repetitive RNAs that (1) can be translated generating five different arginine-rich dipeptide repeat (DPR) proteins that have been shown to alter nucleocytoplasmic transport impacting neuronal survival or (2) not translated causing the accumulation of RNA in RNA foci forming secondary structures which affect promoter activity, genetic instability, RNA splicing, localization, and transport system. Created with BioRender.

**Table 1 jpm-13-01396-t001:** Clinical and diagnostic test characteristics of the studied symptomatic patients.

	II.1	II.5	II.6
Age at diagnosis (years)	88	80	51
Sex (m:f)	f	m	f
Initial symptom	language difficulty	behavioral changes	dysarthria
		cognitive decline	
UMN signs	−	−	+
LMN signs	−	−	+
Dementia	+	+	−
Neuroimaging (MRI)	normal	bilateral frontotemporal cortical atrophy	−
EMG	−	−	Neurogenic changes
Final diagnosis	nfvPPA	bvFTD	ALS

Abbreviations: ALS, amyotrophic lateral sclerosis; bvFTD, behavioral variant frontotemporal dementia; EMG, needle electromyography; LMN, lower motor neuron; MRI, magnetic resonance imaging; nfvPPA, non-fluent variants of primary progressive aphasia; UMN, upper motor neuron.

## Data Availability

No new data were created or analyzed in this study. Data sharing is not applicable to this article.

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
