# Peer review of "C9ORF72 Gene GGGGCC Hexanucleotide Expansion: A High Clinical Variability from Amyotrophic Lateral Sclerosis to Frontotemporal Dementia"

_jpm, 2023, doi:10.3390/jpm13091396_

Round 1
Reviewer 1 Report
This is an impressive manuscript, comprehensive and articulate. It taught me a lot. One question: might you be willing to speculate why so few members of the second generation expressed the expansion?
Author Response
The reviewer's contribution on penetrance was very interesting.
The following paragraph has been added to the article in order to complete the suggested information:
Although the inheritance associated with C9ORF72 is autosomal dominant and, therefore, each descendant of a carrier has a 50% chance of inheriting the expansion, in this family, the percentage of carriers is lower than would be expected a priori. A few years ago, French researchers had already suggested the possibility that gender might influence the transmission of the expansion as, in their study, it was more frequent when the carriers were the mothers than when the fathers were the carriers [35]. In this family, the carriers of the expansion are males. Perhaps this may explain the distortion of transmission.
Reviewer 2 Report
This study reports the pathological significance of of C9ORF72-HRE, common genetic cause of familial forms of ALS, FTD, and concurrent ALS and FTD. This study was performed using human blood/DNA samples. Despite the limitation of human patients samples, the results of this study could provide the readers with the knowledge regarding the relationship by C9ORF72-HRE in ALS and FTD.
[Points]
1. In the text, the information of human patients should be listed as a Table.
Author Response
As suggested by the reviewer, a summary table of the clinical and diagnostic characteristics of the patients evaluated has been added.